# Country and Gender Differences in the Color Association with Energy Drinks: A Survey in Taiwanese and Japanese Students

**DOI:** 10.3390/foods9111670

**Published:** 2020-11-15

**Authors:** Shigeharu Tanei, Wen-Tseng Chu, Toshimitsu Okamura, Fu-Shih Chen, Yukinori Nagakura

**Affiliations:** 1Faculty of Pharmaceutical Sciences, Nihon Pharmaceutical University, 10281 Komuro, Ina-machi, Kitaadachi-gun, Saitama 362-0806, Japan; s-tanei@nichiyaku.ac.jp (S.T.); f-chen@nichiyaku.ac.jp (F.-S.C.); 2Faculty of Graduate Institute of Sport, Leisure, and Hospitality Management, National Taiwan Normal University, No. 162 Heping E. Road, Dah-An District, Taipei 10610, Taiwan; pikachuj@ntnu.edu.tw; 3Faculty of Education for Human Growth, Naragakuen University, 3-12-1 Tatsunokita, Sango-cho, Ikoma-gun, Nara 636-8503, Japan; okamura@naragakuen-u.jp; 4School of Pharmacy at Fukuoka, International University of Health and Welfare, 137-1 Enokizu, Okawa-city, Fukuoka 831-8501, Japan

**Keywords:** energy drinks, color association, inter-country difference, inter-gender difference, Taiwan, Japan

## Abstract

This study investigated differences in the color association with energy drinks between two populations in different cultures, i.e., Taiwanese and Japanese. An anonymous, self-administered paper questionnaire was administered to first- and second-year students at National Taiwan Normal University (Taiwan) and Naragakuen University (Japan). In our inter-country, gender-stratified comparison, the color selected most often in response to the question, “What color comes to your mind for energy drink label?” was red for the Taiwanese and blue for the Japanese. The color associations with energy drinks selected by 20% or more participants in at least one population and showing statistical difference were extracted as noticeable difference. The present study demonstrates that the color and energy drink functions are closely associated. Specifically, yellow and nourishment, black and stimulant, yellow and vitamin supplement, green and dietary fiber supplement, and red and iron supplement are tightly associated regardless of the country. The strong tie between cosmetic and white is specific to the Taiwanese consumers. This suggests that careful color selection based on consumers’ environmental and cultural backgrounds is important in communicating information regarding energy drink functions. It would be worth for energy drink manufacturers to consider those associations in designing labels for products.

## 1. Introduction

Energy drinks are easily available to consumers at various stores. The consumption of energy drinks is huge and rapidly increasing around the world [1]. These drinks contain various ingredients that are advertised as having functions beneficial to the body, such as nourishment, stimulant, cosmetic, vitamin supplement, dietary fiber supplement, and iron supplement (anti-anemic) functions. For example, the ingredient taurine is considered to have a nourishing function because it is involved in various physiological functions, including enzyme activity regulation [2]. Caffeine works as a stimulant to help keep one awake and alert by its stimulatory effect on the central nervous system [3]. Vitamin B2 (riboflavin), one of the B-vitamins, acts as a coenzyme that is involved in redox reactions of nutrients [4]. Collagen is an extracellular matrix protein in connective tissues including skin and joints, and has attracted great interest in the food and cosmetics field [5,6]. Dietary fiber is thought to have beneficial effects on various bodily conditions including metabolic syndrome and gastrointestinal motor activity [7]. Iron is often used to treat anemia associated with iron deficiency [8].

Since the ingredients of energy drinks affect various bodily functions, consumers should select and ingest energy drinks based on an understanding of the effects of the ingredients. However, energy drinks are mainly consumed by adolescents and young adults who do not have sufficient knowledge regarding the effects of the ingredients [9,10]. A survey of college students in the US demonstrated that more than 50% of the respondents had consumed one or more energy drink(s) during the previous month [11]. Another survey of energy drink consumption among adolescents indicated that energy drinks were consumed by 30% to 50% of the global respondents [12]. Labeling of food products has roles in informing consumers of product properties and helping them choose the beneficial (e.g., healthy) products [13,14]. While food labels are used to convey food product′s nutritional properties to consumers, their color substantially influences consumer’s perceptions of a product′s properties. For example, a candy bar wrapped in a green label looked healthier than that in red label [15]. Indeed, adolescents consume energy drinks by picturing their effect on the body based on visual cues such as label color, without considering ingredients [16]. Namely, color works as a perceptual stimulus and has an impact on consumer’s psychology and behavior including beverage products consumption [17]. Given that descriptions on the energy drink label do not effectively communicate information about the effects of the ingredients, the label color is an important means for conveying such information to consumers, especially adolescents.

The purpose of this study was to survey the color association with energy drinks (that is, how various colors used on labels influence consumers’ psychology in consuming energy drinks) among late adolescents. Since color preferences may vary substantially based on the environmental and cultural backgrounds [18,19], the survey was conducted in two East Asian island countries, Taiwan and Japan. Examples of Taiwanese major energy drink products include Vidal (Nanya Food Industry Co., Taoyuan, Taiwan), Comebest (Grape King Biotechnology Co., Taoyuan, Taiwan), Ma Lihan (True Taste Food co., Kaohsiung, Taiwan), and Big husband (Julin Biochemical Technology Co., Taipei, Taiwan). These Taiwanese products predominantly employ red color in their labels. On the other hand, Lipovitan D (Taisho Pharma Co., Tokyo, Japan), Oronamine C (Otsuka Pharmaceutical Co., Tokyo, Japan), and Tiovita (Taiho Pharma. Co., Tokyo, Japan) are included as examples of Japanese popular products. Lipovitan D and Tiovita use blue as the dominant color in the label, whereas Oronamine C employs red. The amounts of predominant ingredients in these products are vitamin C 20 mg, vitamin B2 0.75 mg, vitamin B1 0.06 mg for Vidal (330 mL); caffeine 20 mg, cocktail of vitamins for Comebest (160 mL); inositol 13 mg, nicotinamide 8 mg, cocktail of vitamins for Ma Lihan (150 mL); taurine 1000 mg, caffeine 50 mg, inositol 50 mg, nicotinamide 20 mg, cocktail of vitamins for Lipovitan D (100 mL) and Tiovita (100 mL); and vitamin C 220 mg, caffein 18 mg, niacin 12 mg, vitamin B6 5 mg for Oronamine C (120 mL). The energy drink advertising is being conducted in a similar way between Taiwan and Japan. Although TV commercials are primarily used, web advertising, advertisement using SNS, and billboard advertising are also used. These advertisements give consumers the impression that the energy drinks increase mental and physical energy by using keywords such as energy, vitality, nutrition supply, and fatigue recovery. For example, in the TV commercial for Lipovitan D, popular celebrities say “Fight!” in a powerful voice by holding the product.

## 2. Materials and Methods

### 2.1. Study Design and Respondents

An anonymous, voluntary, and self-administered paper questionnaire was administered to demographically matched late adolescents [20,21]. First- and second-year students enrolled in the liberal arts subject (named General Literacy) at National Taiwan Normal University (Taipei, Taiwan), and first- and second-year students enrolled in the subject (named Educational Psychology) at Naragakuen University (Nara, Japan) were unconditionally included as target population for the survey. The Taiwanese students belonged to the faculties of Education, Literature, Mathematics, Art, Technology, Sport and Leisure, International Social Science, Music or Management Science, while the Japanese students belonged to the faculty of Education for Human Growth. As these faculties were not related to Medical Sciences, Nutrition, or Food science, the students (respondents) were not studying the academic discipline fields which were directly related to the theme of the present study (i.e., color and energy drink). The students had never taken subjects which were directly related to the theme of the present study. There were not students of foreign nationality among the respondents. The paper questionnaires were distributed to a total of 277 Taiwanese students (128 males, 149 females) and 149 Japanese students (69 males, 80 females).

The survey was composed of two sections, student demographics and questions regarding color association with energy drinks. Student demographics included age, height, body weight, and gender. The questions regarding color association with energy drinks were as follows.

Q1. What color is eye-catching in the energy drinks section?

Q2. What color comes to your mind for an energy drink label?

Q3. What color comes to your mind for nutrients (e.g., taurine)?

Q4. What color comes to your mind for stimulants (e.g., caffeine)?

Q5. What color comes to your mind for cosmetics (e.g., collagen)?

Q6. What color comes to your mind for vitamins?

Q7. What color comes to your mind for dietary fibers?

Q8. What color comes to your mind for iron?

In terms of response options, respondents were asked to select one color they visualized in their mind from 13 options (i.e., red, orange, yellow, green, blue, indigo, purple, white, black, gold, silver, pink, and other color). If they selected the “other color” option, they were asked to specify it. The questionnaires were translated into Chinese and Japanese for Taiwanese and Japanese students, respectively. Verbal and written explanations about the survey were given to the students. They were informed about the purpose of the survey and that participation in the survey was voluntary. It was explained to the students that they will not suffer any disadvantage for refusing to participate in the survey or withdrawing their permission at a later stage.

### 2.2. Statistical Analyses

EZR software version 1.40 (Saitama Medical Center, Jichi Medical University, Saitama, Japan) was used for Fisher′s exact test [22]. BellCurve for Excel version 3.20 (Social Survey Research Information Co., Ltd., Tokyo, Japan) was used for other statistical analyses. Descriptive statistics were used to summarize demographic characteristics of the respondents. Categorical variables were presented as the frequency (the number of respondents in a given category) and percentage frequency (the frequency in each category was divided by the total number of corresponding respondents and multiplied by 100), and continuous variables as the mean and standard deviation. Differences in age, height, and body weight between Taiwanese and Japanese students were assessed using Student’s unpaired t-test. The comparison regarding the color association with energy drinks between countries by gender and between genders by country was conducted using Fisher′s exact test. P values less than 0.05 were considered statistically significant. The color associations with energy drinks selected by 20% or more respondents in at least one population and showing statistical difference in inter-country or inter-gender comparison were extracted as exhibiting noticeable difference.

## 3. Results

Valid responses were obtained from 125 Taiwanese students (64 males [M], 61 females [F]) and 105 Japanese students (42 M, 63 F). The valid response rate was 45.1% for Taiwanese students and 70.5% for Japanese students. Demographic characteristics of the valid respondents were summarized by country and gender (Table 1). There was no statistical difference regarding age, height, or body weight between Taiwanese and Japanese respondents.

The average of percentage frequency ([the number of respondents who selected each color]·100/[the total number of corresponding respondents]) in the questionnaire regarding color association with energy drinks is represented by country and gender in Figure 1. The results in inter-country gender-stratified and inter-gender country-stratified comparisons by Fisher’s exact test are presented in Table 2 and Table 3, respectively. In the inter-country gender-stratified comparison, “color that comes to mind for an energy drink label” (answer to Q 2) was significantly different between the countries; that is, Taiwanese respondents selected red color most frequently (27% M, 48% F) and Japanese respondents selected blue (26% M, 35% F), as shown in Figure 1B.

Color association with nourishment (yellow: Taiwanese > Japanese in both genders), cosmetics (white: Taiwanese > J in both genders, pink: Taiwanese < Japanese in both genders), vitamin supplements (yellow: Taiwanese < Japanese in both genders), and iron supplements (black: Taiwanese > Japanese in M, purple: Taiwanese < Japanese in F) were extracted as noticeable differences, as shown in Table 2.

The inter-gender country-stratified comparison showed that color association with stimulants (red: M < F in Taiwanese), dietary fiber supplements (green: M < F in Taiwanese), and iron supplements (red: M < F in Taiwanese, silver: M > F in Taiwanese, purple: M < F in Japanese) met the criteria of noticeable difference as shown in Table 3.

## 4. Discussion

This study administered a questionnaire to late adolescents, i.e., Taiwanese and Japanese university students. In the inter-country gender-stratified comparison, findings regarding the color association with energy drinks were as follows: The result for Q1 suggests that eye-catching colors in the energy drinks section are mainly red and yellow, regardless of country. Color psychology research has suggested that different colors have different effects on the psychology and behaviors of humans; for example, red increases attention and aggressiveness while blue increases subjective alertness [23,24]. It is noteworthy that some associations between color and psychology are limited to specific populations; for example, red was associated with enthusiasm only in a Chinese population [24]. Energy drink distributors might consider that red and yellow are appropriate colors for energy drinks to attract the attention of both Taiwanese and Japanese consumers. The color selected most frequently in response to Q2 (colors which come to mind for an energy drink label) were red (Taiwanese sample) and blue (Japanese sample). It is worth noting that very few Taiwanese respondents selected blue. One possible interpretation for the difference is that label colors used for popular energy drinks in each country considerably influenced the psychology in choosing answer options. According to the ecological valence theory, people generally favor a color in proportion to the degree with which they favor an object associated with that color [25]. Presumably, red and blue are predominantly used in the labels of popular energy drinks in Taiwan and Japan, respectively. Indeed, popular products in Taiwan, e.g., Vidal, Comebest, Ma Lihan, and Big husband, predominantly employ red color in their labels. Major energy drink products in Japan, e.g., Lipovitan D and Tiovita, use blue as the dominant color in the label. The color selected the most in response to Q3 (color association with nutrients) was yellow (around 70% in Taiwan and 30% in Japan) and the percentage frequencies for other colors was less than 20% in both countries. Likewise, the color selected the most in response to Q4 (color association with stimulants) was black in both countries. These results demonstrate that the color association with energy drink functions, i.e., “yellow with nourishments” and “black with stimulants”, have been established in both Taiwan and Japan. In the response to Q5 (color association with cosmetics), a substantial difference between Taiwanese and Japanese respondents was found. Taiwanese preferred white the most by far; however less than 20% of Japanese preferred it, and most selected pink. This would suggest that the particularly strong tie between cosmetic functions and white is specific to the Taiwanese consumers. Based on this result, different colors would be needed to communicate the cosmetic functions of energy drinks to consumers in Taiwan and Japan. In response to Q6 (color association with vitamin supplements), yellow was most commonly selected by Japanese respondents, but the answers were equally dispersed between yellow and green (around 30% each) among Taiwanese respondents. In response to Q7 (color association with dietary fiber supplements), the most frequently selected color by far was green in both countries. This result demonstrates a strong tie between the dietary fiber supplement function and green regardless of the country. One possible interpretation for this is that dietary fiber is contained in edible plant foods including vegetables, fruit, and grains [26]. Green is recognized as the dominant color for plant leaves and stems. In response to Q8 (color association with iron supplements), both Taiwanese and Japanese respondents selected red most. This strong association is possibly due to the recognition that iron supplementation treats iron deficiency anemia that is a decrease in red blood cells [27]. Respondents, except for the female population, frequently selected silver as well, possibly because it is a common color for ironware. Based on the results of the present study, the color association with energy drink functions is common for some combinations (e.g., green and dietary fiber supplements). There are, however, significant differences between the Taiwanese and Japanese respondents for several combinations (e.g., color association with cosmetics) possibly due to environmental and cultural differences between the two countries. The difference is consistent with a previous comparative study among Japan, China, and Indonesia, which demonstrated that environmental and cultural differences were associated with unique color preferences [19].

The present study also conducted an inter-gender country-stratified comparison and obtained the following findings. The color association with energy drinks was generally consistent between males and females in Japan. On the other hand, significant differences were found between genders for several combinations in Taiwan. For example, although few Taiwanese males selected red in response to Q4 (color association with stimulants), around 20% of females selected it. Furthermore, in response to Q8 (color association with iron supplements), Taiwanese females selected red significantly more frequently than males, who, in turn, selected silver more frequently than females. This difference between genders could be because iron deficiency anemia is more common in females **[28]** and females are more aware of iron supplementation to treat anemia. The differences between genders regarding psychology in choosing color are consistent with a previous study which demonstrated that color preference significantly varies depending on gender, age, body mass index, and education in Finnish adult consumers **[18]**.

Energy drinks are very popular, especially among young people, due to the advertisement that they increase physical and mental performance. It should, however, be noted that their intake is associated with various health risks. Accumulating evidence suggest that excessive intake of ingredients of energy drink, e.g., caffeine and taurine, causes adverse events including risk-seeking behaviors, alterations in the cardiovascular system, and poor mental health [29]. The intake of energy drinks in combination with alcohol increases the risk [30,31]. It is important to appreciate that the beneficial and adverse effects of energy drink on consumers are dependent on the frequency of energy drink intake. For example, a recent study showed that the daily consumers perceived more beneficial effects and suffered more adverse effects compared to those who ingest energy drinks less frequently [32]. Furthermore, repeated intake of caffeinated energy drinks could cause the development of dependence to caffeine [32,33], and caffeine-dependent persons are reinforced to ingest caffeinated energy drinks more and vulnerable to adverse events. Due to such health risks to young people, regulatory authorities of individual countries call for stricter regulations on energy drinks. Regulations for daily intake of caffeine, one of representative ingredients of energy drink, is in progress around the world [34,35,36]. Taken together, it is critical to manage the health risks related to the consumption of energy drinks.

One of the strengths of this study is associated with respondents’ demographics; since the purpose of the survey was to investigate the color association with energy drinks among late adolescents, and perform inter-country and inter-gender comparison in Taiwan and Japan, we collected answers from respondents who suited the purpose, i.e., university students in Taiwan and Japan. Age, weight, and height of the respondents were matched precisely between cohorts for comparison. Furthermore, gender-stratified and country-stratified comparisons were conducted to control for confounding variables. On the other hand, this also imposed a limitation on the study: there remains a possibility that results in the present study were influenced by possible sampling biases (e.g., college year and faculties of the students). Thus, it remains to be verified whether the findings in this study can be extrapolated to wider populations. In addition, there may be variables other than nationality and gender, which significantly affect the color choice of energy drink consumers. For example, the frequency of consuming energy drinks should be considered to be an important variable, because a recent study showed that it was negatively correlated with consumer’s knowledge about energy drinks [32]. The analysis by focusing on such variables would be warranted. Furthermore, beverages are divided into several categories, and it might be difficult for consumers to distinguish perfectly between energy drinks and other category of beverages such as sports drinks. The survey on the ability of consumers to discriminate the category of beverages would be considered in the future research.

## 5. Conclusions

The present study demonstrates that the color and energy drink functions are closely associated. Specifically, yellow and nourishment, black and stimulant, yellow and vitamin supplement, green and dietary fiber supplement, and red and iron supplement are tightly associated regardless of the country. The strong tie between cosmetic and white is specific to the Taiwanese consumers. This suggests that careful color selection based on consumers’ environmental and cultural backgrounds is important in communicating information regarding energy drink functions. It would be worth for energy drink manufacturers to consider those associations in designing labels for products.

## Figures and Tables

**Figure 1 foods-09-01670-f001:**
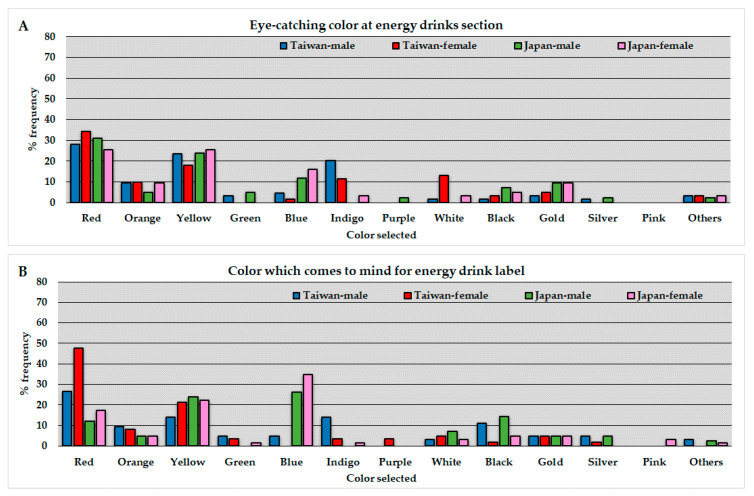
Distributions of respondents on colors in the energy drink-associated questionnaires; (**A**) eye-catching color in the energy drinks section, (**B**) color which comes to mind for an energy drink label, (**C**) color associated with nutrients, (**D**) color associated with stimulants, (**E**) color associated with cosmetic, (**F**) color associated with vitamins, (**G**) color associated with dietary fibers, and (**H**) color associated with iron. Each column represents the average of percentage frequency ([the number of respondents who selected each color] × 100/[the total number of corresponding respondents]) for the four groups (Taiwan-Male, Taiwan-Female, Japan-Male, Japan-Female).

**Table 1 foods-09-01670-t001:** Demographic characteristics of the respondents (Taiwanese and Japanese students).

Variables	Taiwanese	Japanese	*P* Value
Total number	125	105	
Male	64 (51.2%)	42 (40.0%)	
Female	61 (48.8%)	63 (60.0%)	
Age (years old)			
Male	20.2 ± 2.0	19.7 ± 1.3	0.119
Female	20.3 ± 1.6	19.6 ± 2.2	0.05
Height (cm)			
Male	173.5 ± 5.6	173.7 ± 6.4	0.864
Female	160.8 ± 4.7	158.9 ± 6.1	0.055
Weight (kg)			
Male	67.7± 12.3	65.4 ± 9.7	0.316
Female	53.4 ± 7.5	51.4 ± 6.4	0.142

Values are represented as number (%) or mean ± standard deviation.

**Table 2 foods-09-01670-t002:** Inter-country comparison by Fisher′s exact test.

	Male	Female
Q1 (Eye catching colors)	☆ Indigo; *P* = 0.001, T ^#^:20%, J ^##^:0%	Blue; *P* = 0.009, T:2%, J:16%
Q2 (Colors for Label)	☆ Blue: *P* = 0.002, T:5%, J:26% Indigo; *P* = 0.001, T:14%, J:0%	☆ Red; *P* = 0.001, T:48%, J:17% ☆ Blue; *P* < 0.001, T:0%, J:35%
Q3 (Nutrients)	☆ Yellow; *P* = 0.005, T:64%, J:36%	Red; *P* = 0.030, T:3%, J:16% ☆ Yellow; *P* = 0.001, T:75%, J:33% Blue; *P* = 0.028, T:0%, J:10%
Q4 (Stimulants)	Blue; *P* <0.001, T:0%, J:19% Others; *P* = 0.047, T:16%, J:2%	Blue; *P* = 0.006, T:0%, J:13%
Q5 (Cosmetics)	Blue; *P* = 0.023, T:0%, J:10% ☆ White; *P* < 0.001, T:67%, J:19% ☆ Pink; *P* < 0.001, T:0%, J:21%	Red; *P* = 0.003, T:0%, J:14% ☆ White; *P* = 0.001, T:70%, J:14% ☆ Pink; *P* = 0.001, T:15%, J:38%
Q6 (Vitamins)	☆ Yellow; *P* < 0.001, T:30%, J:64%	☆ Yellow; *P* = 0.001, T:44%, J:75%
Q7 (Dietary fibers)	☆ Green; *P* = 0.015, T:48%, J:74% White; *P* = 0.006, T:16%, J:0%	
Q8 (Iron)	☆ Black; *P* = 0.008, T:20%, J:2%	☆ Purple; *P* = 0.016, T:8%, J:25% Black; *P* = 0.002, T:18%, J:2% ☆ Silver; *P* = 0.020, T:7%, J:22%

Results in inter-country comparison using Fisher′s exact test were summarized. The color associations with energy drinks, which showed statistical difference, were extracted and *P* values were presented. The average values of percentage frequency ([the number of respondents who selected each color]100/[the total number of corresponding respondents]) were also shown. The color associations with energy drinks selected by 20% or more respondents in at least one population and showing statistical difference in inter-country comparison were marked with “☆” as noticeable difference. ^#^: Taiwanese, ^##^: Japanese.

**Table 3 foods-09-01670-t003:** Inter-gender comparison by Fisher′s exact test.

	Taiwanese	Japanese
Q1 (Eye catching colors)	White; *P* = 0.015, M ^#^:2%, F ^##^:13%	
Q2 (Colors for Label)	☆ Red; *P* = 0.017, M:27%, F:48%	
Q3 (Nutrients)		
Q4 (Stimulants)	☆ Red; *P* = 0.013, M:5%, F:20%	
Q5 (Cosmetics)	Red; *P* = 0.006, M:13%, F:0% Pink; *P* = 0.001, M:0%, F:15%	
Q6 (Vitamins)		
Q7 (Dietary fibers)	☆ Green; *P* = 0.010, M:48%, F:72% White; *P* = 0.030, M:16%, F:3%	
Q8 (Iron)	☆ Red; *P* = 0.012, M:31%, F:54% ☆ Silver; *P* = 0.004, M:27%, F:7%	☆ Purple; *P* = 0.047, M:10%, F:25%

Results in inter-gender comparison using Fisher′s exact test were summarized. The color associations with energy drinks, which showed statistical difference, were extracted and *P* values were presented in the box. The average values of percentage frequency ([the number of respondents who selected each color] × 100/[the total number of corresponding respondents]) were also shown. The color associations with energy drinks selected by 20% or more respondents in at least one population and showing statistical difference in inter-gender comparison were marked with “☆” as noticeable difference. ^#^: Taiwanese, ^##^: Japanese.

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
