# Peer review of "Country and Gender Differences in the Color Association with Energy Drinks: A Survey in Taiwanese and Japanese Students"

_foods, 2020, doi:10.3390/foods9111670_

Round 1

Reviewer 1 Report

This is a good-quality paper, addressing an important research question with adequate methodology. International comparative research was conducted to analyse colour associations with energy drinks among university students in Japan and Taiwan.

In the title, I would replace "connection to" with "association with".

Were the students in Japan and Taiwan following the same field of studies? How the field of studies could affect your results?

What are the prevailing colours used for energy drinks on the Japanese and Taiwanese markets? You provide some information with the "presumably" qualifier. Please check the actual data.

I think the frequency of consuming energy drinks may be also an important variable in your study. It would be interesting to compare your results between regular users, occasional users, and non-users of energy drinks.

Can 20-year-old people still be called adolescents?

line 174 - distributors

191 - the Taiwanese consumers

198 - the country

199 - vegetables

Author Response

Reviewer 1

[Reviewer’s comment] This is a good-quality paper, addressing an important research question with adequate methodology. International comparative research was conducted to analyse colour associations with energy drinks among university students in Japan and Taiwan.

[Response] We appreciate the thorough evaluation on our manuscript and valuable suggestions from reviewer 1. We have made a thorough review of the points raised and revised the manuscript according to them. The changes in the revised manuscript were shown with blue text.

[Reviewer’s comment] In the title, I would replace "connection to" with "association with".

[Response] Thank you for your suggestion. According to the suggestion, I replaced "connection to" with "association with".

[Reviewer’s comment] Were the students in Japan and Taiwan following the same field of studies? How the field of studies could affect your results?

[Response] Thank you for the important suggestion. The Taiwanese students (respondents) belonged to the faculties of Education, Literature, Mathematics, Art, Technology, Sport and Leisure, International social science, Music or Management science, while the Japanese students belonged to the faculty of Education for Human Growth. As these faculties were not related to Medical Sciences, Nutrition, or Food science, the students were not studying the academic discipline fields which were directly related to the theme of the present study (i.e. color and energy drink). The students had never taken subjects which were directly related to the theme of the present study. We consider it is not likely that the academic discipline fields of the students significantly influenced the results in the present study. According to the suggestion, we clarified the faculties and the subjects in the Materials and Methods as follows (Line 90). “First- and second-year students enrolled in the liberal arts subject (named General Literacy) at National Taiwan Normal University (Taipei, Taiwan), and first- and second-year students enrolled in the subject (named Educational Psychology) at Naragakuen University (Nara, Japan) were unconditionally included as target population for the survey. The Taiwanese students belonged to the faculties of Education, Literature, Mathematics, Art, Technology, Sport and Leisure, International Social Science, Music or Management Science, while the Japanese students belonged to the faculty of Education for Human Growth. As these faculties were not related to Medical Sciences, Nutrition, or Food science, the students (respondents) were not studying the academic discipline fields which were directly related to the theme of the present study (i.e. color and energy drink). The students had never taken subjects which were directly related to the theme of the present study. There were not students of foreign nationality among the respondents.” In addition, we added sentences regarding the future research for verification in the Discussion as follows (Line 276). “In addition, there may be variables other than nationality and gender, which significantly affect the color choice of energy drink consumers. For example, the frequency of consuming energy drinks should be considered as an important variable, because a recent study showed that it was negatively correlated with consumer’s knowledge about energy drinks [32]. The analysis by focusing on such variables would be warranted.”

[Reviewer’s comment] What are the prevailing colours used for energy drinks on the Japanese and Taiwanese markets? You provide some information with the "presumably" qualifier. Please check the actual data. 

[Response] Thank you for the suggestion. We added information about the colors used for popular energy drink products in Taiwan and Japan in the Introduction as follows (Line 69). “Examples of Taiwanese major energy drink products include Vidal (Nanya Food Industry Co., Taoyuan), Comebest (Grape King Biotechnology Co., Taoyuan), Ma Lihan (True Taste Food co., Kaohsiung), and Big husband (Julin Biochemical Technology Co., Taipei). These Taiwanese products predominantly employ red color in their labels. On the other hand, Lipovitan D (Taisho Pharma Co., Tokyo), Oronamine C (Otsuka Pharmaceutical Co., Tokyo), and Tiovita (Taiho Pharma. Co., Tokyo) are included as examples of Japanese popular products. Lipovitan D and Tiovita use blue as the dominant color in the label, whereas Oronamine C employs red.” In addition, the following sentences are added in the Discussion (Line 205). “Presumably, red and blue are predominantly used in the labels of popular energy drinks in Taiwan and Japan, respectively. Indeed, popular products in Taiwan, e.g. Vidal, Comebest, Ma Lihan, and Big husband, predominantly employ red color in their labels. Major energy drink products in Japan, e.g. Lipovitan D and Tiovita, use blue as the dominant color in the label.”

[Reviewer’s comment] I think the frequency of consuming energy drinks may be also an important variable in your study. It would be interesting to compare your results between regular users, occasional users, and non-users of energy drinks.

[Response] Thank you for the insightful suggestion. I agree that frequency of consuming energy drinks is an important variable which may affect the color choice of energy drink consumers. The following sentences are added to the Discussion (Line 276). “In addition, there may be variables other than nationality and gender, which significantly affect the color choice of energy drink consumers. For example, the frequency of consuming energy drinks should be considered as an important variable, because a recent study showed that it was negatively correlated with consumer’s knowledge about energy drinks [32]. The analysis by focusing on such variables would be warranted.”

[Reviewer’s comment] Can 20-year-old people still be called adolescents?

[Response] Thank you for the suggestion. Previous literatures including [20, 21] call the ages around 20 “late adolescence”. According to the suggestion, the respondents (university students) was named “late adolescents”. For example, at line 90, “An anonymous, voluntary, and self-administered paper questionnaire was administered to demographically matched late adolescents [20, 21].”

[Reviewer’s comment] line 174 – distributors

[Response] (Line 198) According to the suggestion, “distributers” was changed to “distributors “.

[Reviewer’s comment] 191 - the Taiwanese consumers

[Response] (Line 218) According to the suggestion, “Taiwanese” was changed to “the Taiwanese consumers “.

[Reviewer’s comment] 198 - the country

[Response] (Line 225) According to the suggestion, “country” was changed to “the country “.

[Reviewer’s comment] 199 – vegetables

[Response] (Line 226) According to the suggestion, “vegetable” was changed to “vegetables “.

Reviewer 2 Report

Comments and Suggestions for Authors

the work is interesting and brings a new perspective on the discussed topic

the topic taken up is very important and needs to be discussed from different points of view

the study was well planned and conducted

English language is good/acceptable in its present form

Font used in the figure 1 should be the same of the text (Palatino Linotype)

some literature on food products and labelling of food products could be added

conclusions must be improved:

the conclusions seem to me too short and generic, not focusing entirely on the study itself, but on conclusions that could be drawn even before the study is carried out. Please highlight the findings of this paper and its importance for the scientific community and for the readers of this journal, moreover tell something about the impact in the economic context, such as producers.

Author Response

Reviewer 2

[Reviewer’s comment] the work is interesting and brings a new perspective on the discussed topic. the topic taken up is very important and needs to be discussed from different points of view. the study was well planned and conducted.

English language is good/acceptable in its present form.

[Response] We appreciate the thorough evaluation on our manuscript and valuable suggestions from reviewer 2. We have made a thorough review of the points raised and revised the manuscript according to them. The changes in the revised manuscript were shown with blue text.

[Reviewer’s comment] Font used in the figure 1 should be the same of the text (Palatino Linotype)

[Response] Thank you for the indication. Font in the Figure 1 was changed to Palatino Linotype.

[Reviewer’s comment] some literature on food products and labelling of food products could be added

[Response] Thank you for the suggestion. Previous literatures, which investigated the food labeling and its effect on consumer’s choice of products, were added as follows (Line 54). “Labelling of food products has roles in informing consumers of product properties and helping them choose the beneficial (e.g. healthy) products [13,14]. While food labels are used to convey food product's nutritional properties to consumers, their color substantially influences consumer’s perceptions of a product's properties. For example, a candy bar wrapped in green label looked healthier than that in red label [15].”  

[Reviewer’s comment] conclusions must be improved: the conclusions seem to me too short and generic, not focusing entirely on the study itself, but on conclusions that could be drawn even before the study is carried out. Please highlight the findings of this paper and its importance for the scientific community and for the readers of this journal, moreover tell something about the impact in the economic context, such as producers.

[Response] We appreciate your suggestion. According to the suggestion, I considerably rectified the conclusion by focusing on the present study as follows. (Abstract and Line 286) “The present study demonstrates that the color and energy drink functions are closely associated. Specifically, yellow and nourishment, black and stimulant, yellow and vitamin supplement, green and dietary fiber supplement, and red and iron supplement are tightly associated regardless of the country. The strong tie between cosmetic and white is specific to the Taiwanese consumers. This suggests that careful color selection based on consumers’ environmental and cultural backgrounds is important in communicating information regarding energy drink functions. It would be worth for energy drink manufacturers to consider those associations in designing labels for products.”

Reviewer 3 Report

This paper discusses the color connection to energy drinks among Taiwanese and Japanese students in their first- and second-year of university education. The main contribution of this paper is that it further reinforces the idea, that the color connection to energy drink function significantly varies depending on the country and gender in our “global World”. Color connection to nourishment and cosmetics is an interesting topic and worth to study in different products and countries.

Major comments

  1. The discussion about the volume and age limit and the acceptable daily intake (ADI) of energy drink consumption would be useful to mention. I admit, this is not the objective of the paper, but it worth some sentences, maybe among the limitations of the study. Too much herbal stimulants, caffeine, sugar and taurine can cause harm (Al-Shaar et al., 2017; Beacon Health System, 2020). A glimpse of the literature displayed that energy drinks possesses a plethora of impressive healthful benefits along with numerous health risks (Kaur et al., 2019). See more about the health effects of caffeine in Temple et al, 2017.

In case reports, high consumption of energy drinks, especially when mixed with alcohol, has been linked to adverse cardiovascular, psychological, and neurologic events, including fatal events (Ehlers et al., 2019). Consumption of energy drinks in combination with alcohol should be avoided (van Dam et al., 2020).

  1. Also, the psychology of colors is important in case of labelling, but also in case of advertisement. Some sentences about the energy drink advertising situation in Taiwan and Japan (e. g. the Lipovitan brand sold in small brown glass medicine bottles) would help a lot to a reader far away from these countries. Specially, children and adolescents are attracted to energy drinks not only due to effective marketing, but to influence from peers, and lack of knowledge about their potential harmful effects (Pound - Blair, 2017).

Also, in a study in the Kingdom of Saudi Arabia, the total score of perceived beneficial effects about energy drinks (involving feeling mood elevation and increase in vitality/energy, helping with athletic/academic performance, feeling an increase in focus/memory and helping with long-distance driving) was the highest among the daily consumers compared to the rest of the consumption pattern groups. It was despite that the daily users experienced significantly more adverse effects than the other three consuming groups consuming less. Also, tobacco smoking determined consumption patterns as the majority of daily energy drink consumers were smokers (Subaiea et al., 2019). These findings are consistent with the findings of other studies that aimed to determine the associated factors with energy drink consumption worldwide (Miller, 2008; Attila - Cakir, 2011).

It is important to mention that studies have shown that caffeinated energy drinks can cause addiction and dependence (Reissig et al., 2009). When caffeine containing energy drinks are taken in high amounts, they induce arousal by caffeine’s effect on the pleasure area of brain, which is similar to the effects of tobacco, alcohol, and other drugs that should be treated with caution to avoid abuse (Persad, 2011).

These explains a complex situation where the psychology of colors is just one factor.

  1. Sports and energy drinks are advertised to appeal to those who exercise or need a boost of energy to get through the day. The “angel wings” advertisement of Red Bull, including red, blue, silver and some yellow colors is a good example. But sports beverages and energy drinks are a different product entirely. Sport drinks are devoid of any kind of stimulants (Kaur et al., 2019). Most of the sports drinks consist of low percentage of carbohydrate along with minerals like sodium and potassium, whereas energy drinks have a high carbohydrate concentration (Koester, 2009 cited by Kaur et al., 2019). Additionally, Red Bull has sugar-free options, including Red Bull Zero and Red Bull Sugarfree, which are made with the artificial sweeteners aspartame and acesulfame K instead of sugar (Miles-Chan et al., 2015). Did the respondents know the difference between sport and energy drinks? People are sometimes confused with these categories.

Additionally, a pilot study with the survey to verify content clarity and accuracy is suggested in the next study.

Minor comments

  1. Procedures in this research were approved by the Ethical Committee of Nihon Pharmaceutical University (Permission number: 1-10) located in Japan. It seems to be useful to set up ethical committee in the universities taking part in this research if they want to continue this work.
  2. Also, were there foreign students among the respondents? Are there foreign students on the campus of the two universities? What was the nationality of them, if they were involved? Foreign students can affect the food and drink consumption pattern of local students they meet.
  3. It would be useful to involve the rate of energy drink consumption of the given respondent in the next study. Loyalty can be an important and interesting factor, parallel with the nationality and gender. See the recent study in the Kingdom of Saudi Arabia (Subaiea et al., 2019) for example.

Questions

  1. What kind of energy drinks are available in Taiwan and Japan? What colors do they use on their labels, bottles and what is the actual color of the drink? It is mentioned in the paper that “red and blue are predominantly used in the labels of popular energy drinks in Taiwan and Japan, respectively”.

What do the energy drinks contain available in these countries? Most of these kinds of drinks contain nearly 80 mg of caffeine per serving, equaling almost eight ounces of coffee, whereas some may even supply approximately 300 mg caffeine per serving (Reid et al., 2017). Many energy drinks pack about 200 mg of caffeine, the amount in two cups of brewed coffee. Researchers claim that changes in heart rhythm can’t be explained solely by the amount of caffeine, but are likely due to the combination of ingredients in Red Bull (Shah et al., 2019). Most of the readers of this paper are not familiar with the energy drink market in Taiwan and Japan. A little more information would be helpful.

The European Food Safety Authority has proposed a conservative safety level of 3 milligrams of caffeine per kilogram of body weight (mg/kg bw) in a single dose for children and adolescents (EFSA, 2015). The threshold of caffeine toxicity appears to be around 400 mg/day in healthy adults (19 years or older), 100 mg/day in healthy adolescents (12–18 years old), and 2.5 mg/kg/day in healthy children (less than 12 years old) (Seifert et al., 2011). For comparison, one standard sized can of a popular energy drink provides 77 mg of caffeine (or 1.1 mg/kg/day) for a 70-kg male and twice that, 2.2 mg/kg/day, for a 35-kg pre-teen (Oddy - O'Sullivan, 2009). The total amount of caffeine contained in some energy drinks can exceed 500 mg (equivalent to 14 cans of common caffeinated soft drinks or 5 cups of coffee) and is high enough to be toxic in children and young adults (Reissig et al., 2009). The caffeine content in soft drinks is regulated by the US Food and Drug Administration but not in energy drinks, as they are declared safe under the US Code of Federal Regulations. The American Academy of Pediatrics’ Committee on Nutrition and the Council on Sports Medicine and Fitness recently concluded that “rigorous review and analysis of the literature reveal that caffeine and other stimulant substances contained in energy drinks have no place in the diet of children and adolescents” (Committee on Nutrition and the Council on Sports Medicine and Fitness, 2011). In the European Union, it becomes imperative to mention “high caffeine content” on the labels of drinks containing more than 150 mg/kg of caffeine (Breda et al., 2014; Peacock et al., 2015). Food Safety and Standards Authority of India has regulated the caffeine addition in energy drinks up to 320 ppm (Bedi et al., 2014). Some energy drink manufacturers bypass the set limits of caffeine (80 mg/250 ml can in the EU) by declaring their product as “dietary supplement” to avoid these limits (Seifert et al., 2011; Breda et al., 2014). Moreover, energy drinks are regarded as dietary supplements in several countries. According to the European Union, Red Bull energy drinks are required to have a “high caffeine content” label, and Canada requires labels indicating that Red Bull should not be mixed with alcohol and that maximum daily consumption should not exceed two 8.3-oz cans (Reissig et al., 2009).

In Australia, the highest recommended intake of caffeine is 160 mg per day and the drinks with more than 320 ppm of caffeine are banned (Peacock et al., 2015). In some countries such as

Denmark, Norway, Uruguay and Turkey, Iceland and France energy drinks have been banned owing to its potent adverse effects (Bedi et al., 2014; Breda et al., 2014) In 2014, a law was introduced to ban the sale of energy drinks containing at least 150 mg caffeine/litre to adolescents less than 18 years in Lithuania. Similarly, the sale of some energy drinks to pharmacies as well as children is banned in Sweden (Breda et al.,2014; Visram - Hashem, 2016).

  1. On what faculty of the National Taiwan Normal University (Taipei, Taiwan) the study was made? What classes the participating students have already passed on these universities? Also, what other faculties are present on the campus of the universities involved?

There can be a major difference between medical students and students learning economics in this case for example.

Also, participating students can meet students from other faculties and get some knowledge this way.

What is more, students involved in education dealing with foods, drinks and health may be worried about telling their honest answer, what is not parallel with what they have learned. The researchers always say that the study participation is anonymous, and voluntary, but students can be afraid that they can be identified or their professors can get nervous in general and this can have consequences on the exams.

Literature

Al-Shaar.  L., Vercammen, K., Lu, C., Richardson, S., Tamez, M., Mattei, J. (2017): Health Effects and Public Health Concerns of Energy Drink Consumption in the United States: A Mini-Review. Frontiers in Public Health, 5:225.

Attila, S., Cakir, B. (2011): Energy-drink consumption in college students and associated factors. Nutrition, 27, 3, 316–322.

Beacon Health System (2020): Taurine in energy drinks: What is it? www.beaconhealthsystem. org/library/faqs/taurine-in-energy-drinks-what-is-it/ (Accessed: 03.11.2020)

Bedi, N., Dewan, P., Gupta, P. (2014): Energy drinks: potions of illusion. Indian Pediatrics, 51, 7, 529-533.

Breda, J. J., Whiting, S. H., Encarnação, R., Norberg, S., Jones, R., Reinap, M., Jewell, J. (2014): Energy drink consumption in Europe: a review of the risks, adverse health effects, and policy options to Respond. Frontiers in Public Health, 2, 134.

Committee on Nutrition and the Council on Sports Medicine and Fitness (2011): Sports drinks and energy drinks for children and adolescents: are they appropriate? Pediatrics 127, 1182–1189.

EFSA (2015): Scientific Opinion on the safety of caffeine. EFSA Journal, 13, 5, 4102.

Ehlers, A., Marakis, G., Lampen, A., Hirsch-Ernst, K. I. (2019): Risk assessment of energy drinks with focus on cardiovascular parameters and energy drink consumption in Europe. Food and Chemical Toxicology, 130, 109-121.

Kaur, J., Kumar, V., Goyal, A., Tanwar, B., Gat, Y., Rasane, P., Suri, S. (2019): Energy drinks: health effects and consumer safety. Nutrition & Food Science. 49, 6, 1075-1087.

Liauchonak, I., Qorri, B., Dawoud, F., Riat, Y., Szewczuk, M. R. (2019): Non-Nutritive Sweeteners and Their Implications on the Development of Metabolic Syndrome. Nutrients, 11, 3, 644.

Miles-Chan, J. L., Charrière, N., Grasser, E. K., Montani, J. P., Dulloo, A. G. (2015): The thermic effect of sugar-free Red Bull: do the non-caffeine bioactive ingredients in energy drinks play a role? Obesity (Silver Spring), 23, 1, 16-19.

Oddy, W. H., O’Sullivan, T. A. (2009): Energy drinks for children and adolescents. BMJ (British Medical Journal), 339 :b5268

Peacock, A., Droste, N., Pennay, A., Miller, P., Lubman, D. I., Bruno, R. (2015): Awareness of energy drink intake guidelines and associated consumption practices: a cross-sectional study. BMC Public Health, 16, 1, 6.

Persad, L. A. B. (2011): Energy drinks and the neurophysiological impact of caffeine. Frontiers in Neuroscience, 5, 116., 1-8.

Pound, C. M., Blair, B. (2017): Canadian Paediatric Society, Nutrition and Gastroenterology Committee, Ottawa, Ontario. Energy and sports drinks in children and adolescents. Paediatrics & Child Health, 22, 7, 406-410.

Reid, J. L., McCrory, C., White, C. M., Martineau, C., Vanderkooy, P., Fenton, N., Hammond, (2017): Consumption of caffeinated energy drinks among youth and young adults in Canada. Preventive Medicine Reports, 5, 65-70.

Reissig, C. J., Strain, E. C., Griffiths, R. R. (2009): Caffeinated energy drinks - a growing problem. Drug and Alcohol Dependence, 99, 1–3), 1–10.

Seifert, S.M., Schaechter, J. L., Hershorin, E. R. Lipshultz, S. E. (2011): Health effects of energy drinks on children, adolescents, and young adults. Pediatrics, 127, 3, 511-528.

Shah, S. A., Szeto, A. H., Farewell, R., Shek, A., Fan, D., Quach, K. N., Bhattacharyya, M., Elmiari, J., Chan, W., O'Dell, K., Nguyen, N., McGaughey, T. J., Nasir, J. M., Kaul, S. (2019): Impact of High Volume Energy Drink Consumption on Electrocardiographic and Blood Pressure Parameters: A Randomized Trial. Journal of the American Heart Association, 8, 11, e011318.

Subaiea, G. M., Altebainawi, A. F., Alshammari, T. M. (2019): Energy drinks and population health: consumption pattern and adverse effects among Saudi population. BMC Public Health, 19, 1539.

Temple, J. L., Bernard, C., Lipshultz, S. E., Czachor, J. D., Westphal, J. A., Mestre, M. A. (2017): The Safety of Ingested Caffeine: A Comprehensive Review. Frontiers in Psychiatry, 8, Article 80.

van Dam, R. M., Hu, F. B., Willett, W. C. (2020): Coffee, Caffeine, and Health. The New England Journal of Medicine, 383, 369-378.

Visram, S., Hashem, K. (2016): Energy drinks: what’s the evidence. Briefing Paper, Food Research Collaboration, London, 1-12., https://foodresearch.org.uk/publications/energy-drinks/

Author Response

Reviewer 3

[Reviewer’s comment] This paper discusses the color connection to energy drinks among Taiwanese and Japanese students in their first- and second-year of university education. The main contribution of this paper is that it further reinforces the idea, that the color connection to energy drink function significantly varies depending on the country and gender in our “global World”. Color connection to nourishment and cosmetics is an interesting topic and worth to study in different products and countries.

[Response] We appreciate the thorough evaluation on our manuscript and valuable suggestions from reviewer 3. We have made a thorough review of the points raised and revised the manuscript according to them. The changes in the revised manuscript were shown with blue text.

Major comments

[Reviewer’s comment] The discussion about the volume and age limit and the acceptable daily intake (ADI) of energy drink consumption would be useful to mention. I admit, this is not the objective of the paper, but it worth some sentences, maybe among the limitations of the study. Too much herbal stimulants, caffeine, sugar and taurine can cause harm (Al-Shaar et al., 2017; Beacon Health System, 2020). A glimpse of the literature displayed that energy drinks possesses a plethora of impressive healthful benefits along with numerous health risks (Kaur et al., 2019). See more about the health effects of caffeine in Temple et al, 2017.

In case reports, high consumption of energy drinks, especially when mixed with alcohol, has been linked to adverse cardiovascular, psychological, and neurologic events, including fatal events (Ehlers et al., 2019). Consumption of energy drinks in combination with alcohol should be avoided (van Dam et al., 2020).

[Response] Thank you for the suggestion. We agree that the health risks related to energy drink intake and its management are important matters to be considered seriously. According to the kind suggestion, we made a paragraph regarding the matters in the Discussion as follows (Line 251). “Energy drinks are very popular, especially among young people, due to the advertisement that they increase physical and mental performances. It should, however, be noted that their intake is associated with various health risks. Accumulating evidence suggest that excessive intake of ingredients of energy drink, e.g. caffeine and taurine, causes adverse events including risk-seeking behaviors, alterations in the cardiovascular system, and poor mental health [29]. The intake of energy drinks in combination with alcohol increases the risk [30,31]. It is important to appreciate that the beneficial and adverse effects of energy drink on consumers are dependent on the frequency of energy drink intake. For example, a recent study showed that the daily consumers perceived more beneficial effects and suffered more adverse effects compared to those who ingest energy drinks less frequently [32]. Furthermore, repeated intake of caffeinated energy drinks could cause the development of dependence to caffeine [32,33], and caffeine-dependent persons are reinforced to ingest caffeinated energy drinks more and vulnerable to adverse events. Due to such health risks to young people, regulatory authorities of individual countries call for stricter regulations on energy drinks. Regulations for daily intake of caffeine, one of representative ingredients of energy drink, is in progress around the world [34-36]. Taken together, it is critical to manage the health risks related to the consumption of energy drinks.”

[Reviewer’s comment] Also, the psychology of colors is important in case of labelling, but also in case of advertisement. Some sentences about the energy drink advertising situation in Taiwan and Japan (e. g. the Lipovitan brand sold in small brown glass medicine bottles) would help a lot to a reader far away from these countries. Specially, children and adolescents are attracted to energy drinks not only due to effective marketing, but to influence from peers, and lack of knowledge about their potential harmful effects (Pound - Blair, 2017).

[Response] Thank you for the suggestion regarding the advertisement of energy drink products in Taiwan and Japan. According to the suggestion, following sentences are added to the Introduction (Line 80). “The energy drink advertising is being conducted in a similar way between Taiwan and Japan. Although TV commercials are primarily used, web advertising, advertisement using SNS, and billboard advertising are also used. These advertisements give consumers the impression that the energy drinks increase mental and physical energy by using keywords such as energy, vitality, nutrition supply, and fatigue recovery. For example, in the TV commercial for Lipovitan D, popular celebrities say “Fight!” in a powerful voice by holding the product.”

[Reviewer’s comment] Also, in a study in the Kingdom of Saudi Arabia, the total score of perceived beneficial effects about energy drinks (involving feeling mood elevation and increase in vitality/energy, helping with athletic/academic performance, feeling an increase in focus/memory and helping with long-distance driving) was the highest among the daily consumers compared to the rest of the consumption pattern groups. It was despite that the daily users experienced significantly more adverse effects than the other three consuming groups consuming less. Also, tobacco smoking determined consumption patterns as the majority of daily energy drink consumers were smokers (Subaiea et al., 2019). These findings are consistent with the findings of other studies that aimed to determine the associated factors with energy drink consumption worldwide (Miller, 2008; Attila - Cakir, 2011).

[Response] Thank you for the information that the beneficial and adverse effects of energy drinks are dependent on the frequency of consumption. According to the suggestion, the following sentences were added to the Discussion (Line 256). “It is important to appreciate that the beneficial and adverse effects of energy drink on consumers are dependent on the frequency of energy drink intake. For example, a recent study showed that the daily consumers perceived more beneficial effects and suffered more adverse effects compared to those who ingest energy drinks less frequently [32].”

[Reviewer’s comment] It is important to mention that studies have shown that caffeinated energy drinks can cause addiction and dependence (Reissig et al., 2009). When caffeine containing energy drinks are taken in high amounts, they induce arousal by caffeine’s effect on the pleasure area of brain, which is similar to the effects of tobacco, alcohol, and other drugs that should be treated with caution to avoid abuse (Persad, 2011).

These explains a complex situation where the psychology of colors is just one factor.

[Response] Thank you for the important information about the risk of dependence caused by caffeine. According to the suggestion, the following sentences were added to the Discussion (Line260). “Furthermore, repeated intake of caffeinated energy drinks could cause the development of dependence to caffeine [32,33], and caffeine-dependent persons are reinforced to ingest caffeinated energy drinks more and vulnerable to adverse events.”

[Reviewer’s comment] Sports and energy drinks are advertised to appeal to those who exercise or need a boost of energy to get through the day. The “angel wings” advertisement of Red Bull, including red, blue, silver and some yellow colors is a good example. But sports beverages and energy drinks are a different product entirely. Sport drinks are devoid of any kind of stimulants (Kaur et al., 2019). Most of the sports drinks consist of low percentage of carbohydrate along with minerals like sodium and potassium, whereas energy drinks have a high carbohydrate concentration (Koester, 2009 cited by Kaur et al., 2019). Additionally, Red Bull has sugar-free options, including Red Bull Zero and Red Bull Sugarfree, which are made with the artificial sweeteners aspartame and acesulfame K instead of sugar (Miles-Chan et al., 2015). Did the respondents know the difference between sport and energy drinks? People are sometimes confused with these categories.

[Reviewer’s comment] Additionally, a pilot study with the survey to verify content clarity and accuracy is suggested in the next study.

[Response] We appreciate the suggestion. Taiwanese and Japanese students (respondents) roughly distinguish between sports and energy drinks by size and shape of the products. For example, Pocari Sweat (Otsuka Pharmaceutical Co., Tokyo), one of major sports drinks, is sold typically in a plastic bottle (250 ml or more), although Lipovitan D (Taisho Pharma Co., Tokyo), one of major energy drinks, is in a small glass bottle (100 ml). However, it might be difficult to perfectly distinguish between sports and energy drinks. According to the suggestion, following sentences were added to the Discussion (Line 281). “Furthermore, beverages are divided into several categories, and it might be difficult for consumers to distinguish perfectly between energy drinks and other category of beverages like sports drinks. The survey on the ability of consumers to discriminate the category of beverages would be considered in the future research.”     

Minor comments

[Reviewer’s comment] Procedures in this research were approved by the Ethical Committee of Nihon Pharmaceutical University (Permission number: 1-10) located in Japan. It seems to be useful to set up ethical committee in the universities taking part in this research if they want to continue this work.

[Response] We appreciate the suggestion which we need to consider in our future research.

[Reviewer’s comment] Also, were there foreign students among the respondents? Are there foreign students on the campus of the two universities? What was the nationality of them, if they were involved? Foreign students can affect the food and drink consumption pattern of local students they meet.

[Response] Thank you for the suggestion. There were not students of foreign nationality among the respondents. We added this information to the Material and Methods as follows (Line 100). “There were not students of foreign nationality among the respondents.”

[Reviewer’s comment] It would be useful to involve the rate of energy drink consumption of the given respondent in the next study. Loyalty can be an important and interesting factor, parallel with the nationality and gender. See the recent study in the Kingdom of Saudi Arabia (Subaiea et al., 2019) for example.

[Response] Thank you for the insightful suggestion. I agree that the rate of energy drink consumption is an important variable which may affect the color choice of energy drink consumers. The following sentences are added to the Discussion (Line 276). “In addition, there may be variables other than nationality and gender, which significantly affect the color choice of energy drink consumers. For example, the frequency of consuming energy drinks should be considered as an important variable, because a recent study showed that it was negatively correlated with consumer’s knowledge about energy drinks [32]. The analysis by focusing on such variables would be warranted.”

Questions

[Reviewer’s comment] What kind of energy drinks are available in Taiwan and Japan? What colors do they use on their labels, bottles and what is the actual color of the drink? It is mentioned in the paper that “red and blue are predominantly used in the labels of popular energy drinks in Taiwan and Japan, respectively”.

[Response] Thank you for the suggestion. We added information about the colors used for popular energy drink products in Taiwan and Japan in the Introduction as follows (Line 69). “Examples of Taiwanese major energy drink products include Vidal (Nanya Food Industry Co., Taoyuan), Comebest (Grape King Biotechnology Co., Taoyuan), Ma Lihan (True Taste Food co., Kaohsiung), and Big husband (Julin Biochemical Technology Co., Taipei). These Taiwanese products predominantly employ red color in their labels. On the other hand, Lipovitan D (Taisho Pharma Co., Tokyo), Oronamine C (Otsuka Pharmaceutical Co., Tokyo), and Tiovita (Taiho Pharma. Co., Tokyo) are included as examples of Japanese popular products. Lipovitan D and Tiovita use blue as the dominant color in the label, whereas Oronamine C employs red.” In addition, the following sentences are added in the Discussion (Line 205). “Presumably, red and blue are predominantly used in the labels of popular energy drinks in Taiwan and Japan, respectively. Indeed, popular products in Taiwan, e.g. Vidal, Comebest, Ma Lihan, and Big husband, predominantly employ red color in their labels. Major energy drink products in Japan, e.g. Lipovitan D and Tiovita, use blue as the dominant color in the label.”

[Reviewer’s comment] What do the energy drinks contain available in these countries? Most of these kinds of drinks contain nearly 80 mg of caffeine per serving, equaling almost eight ounces of coffee, whereas some may even supply approximately 300 mg caffeine per serving (Reid et al., 2017). Many energy drinks pack about 200 mg of caffeine, the amount in two cups of brewed coffee. Researchers claim that changes in heart rhythm can’t be explained solely by the amount of caffeine, but are likely due to the combination of ingredients in Red Bull (Shah et al., 2019). Most of the readers of this paper are not familiar with the energy drink market in Taiwan and Japan. A little more information would be helpful.

[Response] Thank you for your suggestion. The following sentences were added to the Introduction (Line 75). “The amounts of predominant ingredients in these products are vitamin C 20 mg, vitamin B2 0.75 mg, vitamin B1 0.06 mg for Vidal (330 ml); caffeine 20 mg, cocktail of vitamins for Comebest (160 ml); inositol 13 mg, nicotinamide 8 mg, cocktail of vitamins for Ma Lihan (150 ml); taurine 1000 mg, caffeine 50 mg, inositol 50 mg, nicotinamide 20 mg, cocktail of vitamins for Lipovitan D (100 ml) and Tiovita (100 ml); and vitamin C 220 mg, caffein18mg, niacin 12 mg, vitamin B6 5 mg for Oronamine C (120 ml).”

[Reviewer’s comment] The European Food Safety Authority has proposed a conservative safety level of 3 milligrams of caffeine per kilogram of body weight (mg/kg bw) in a single dose for children and adolescents (EFSA, 2015). The threshold of caffeine toxicity appears to be around 400 mg/day in healthy adults (19 years or older), 100 mg/day in healthy adolescents (12–18 years old), and 2.5 mg/kg/day in healthy children (less than 12 years old) (Seifert et al., 2011). For comparison, one standard sized can of a popular energy drink provides 77 mg of caffeine (or 1.1 mg/kg/day) for a 70-kg male and twice that, 2.2 mg/kg/day, for a 35-kg pre-teen (Oddy - O'Sullivan, 2009). The total amount of caffeine contained in some energy drinks can exceed 500 mg (equivalent to 14 cans of common caffeinated soft drinks or 5 cups of coffee) and is high enough to be toxic in children and young adults (Reissig et al., 2009). The caffeine content in soft drinks is regulated by the US Food and Drug Administration but not in energy drinks, as they are declared safe under the US Code of Federal Regulations. The American Academy of Pediatrics’ Committee on Nutrition and the Council on Sports Medicine and Fitness recently concluded that “rigorous review and analysis of the literature reveal that caffeine and other stimulant substances contained in energy drinks have no place in the diet of children and adolescents” (Committee on Nutrition and the Council on Sports Medicine and Fitness, 2011). In the European Union, it becomes imperative to mention “high caffeine content” on the labels of drinks containing more than 150 mg/kg of caffeine (Breda et al., 2014; Peacock et al., 2015). Food Safety and Standards Authority of India has regulated the caffeine addition in energy drinks up to 320 ppm (Bedi et al., 2014). Some energy drink manufacturers bypass the set limits of caffeine (80 mg/250 ml can in the EU) by declaring their product as “dietary supplement” to avoid these limits (Seifert et al., 2011; Breda et al., 2014). Moreover, energy drinks are regarded as dietary supplements in several countries. According to the European Union, Red Bull energy drinks are required to have a “high caffeine content” label, and Canada requires labels indicating that Red Bull should not be mixed with alcohol and that maximum daily consumption should not exceed two 8.3-oz cans (Reissig et al., 2009).

In Australia, the highest recommended intake of caffeine is 160 mg per day and the drinks with more than 320 ppm of caffeine are banned (Peacock et al., 2015). In some countries such as Denmark, Norway, Uruguay and Turkey, Iceland and France energy drinks have been banned owing to its potent adverse effects (Bedi et al., 2014; Breda et al., 2014) In 2014, a law was introduced to ban the sale of energy drinks containing at least 150 mg caffeine/litre to adolescents less than 18 years in Lithuania. Similarly, the sale of some energy drinks to pharmacies as well as children is banned in Sweden (Breda et al.,2014; Visram - Hashem, 2016).

[Response] Thank you for a plenty of information about the regulation of caffein in individual countries. Some of the literatures were cited in the revised manuscript.

[Reviewer’s comment] On what faculty of the National Taiwan Normal University (Taipei, Taiwan) the study was made? What classes the participating students have already passed on these universities? Also, what other faculties are present on the campus of the universities involved?

There can be a major difference between medical students and students learning economics in this case for example.

Also, participating students can meet students from other faculties and get some knowledge this way.

What is more, students involved in education dealing with foods, drinks and health may be worried about telling their honest answer, what is not parallel with what they have learned.

[Response]Thank you for the important suggestion. The Taiwanese students (respondents) belonged to the faculties of Education, Literature, Mathematics, Art, Technology, Sport and Leisure, International social science, Music or Management science, while the Japanese students belonged to the faculty of Education for Human Growth. As these faculties were not related to Medical Sciences, Nutrition, or Food science, the students were not studying the academic discipline fields which were directly related to the theme of the present study (i.e. color and energy drink). The students had never taken subjects which were directly related to the theme of the present study. National Taiwan Normal University (Taipei, Taiwan) has no other faculty. Although Naragakuen University (Nara, Japan) has the Faculty of Health Sciences as the other faculty, it is on another campus. We consider it is not likely that the academic discipline fields of the students significantly influenced the results in the present study. According to the suggestion, we clarified the faculties and the subjects of the students (respondents) in the Materials and Methods as follows (Line 90). “First- and second-year students enrolled in the liberal arts subject (named General Literacy) at National Taiwan Normal University (Taipei, Taiwan), and first- and second-year students enrolled in the subject (named Educational Psychology) at Naragakuen University (Nara, Japan) were unconditionally included as target population for the survey. The Taiwanese students belonged to the faculties of Education, Literature, Mathematics, Art, Technology, Sport and Leisure, International Social Science, Music or Management Science, while the Japanese students belonged to the faculty of Education for Human Growth. As these faculties were not related to Medical Sciences, Nutrition, or Food science, the students (respondents) were not studying the academic discipline fields which were directly related to the theme of the present study (i.e. color and energy drink). The students had never taken subjects which were directly related to the theme of the present study. There were not students of foreign nationality among the respondents.” In addition, we added sentences regarding the future research for verification in the Discussion as follows (Line 274). “there remains a possibility that results in the present study were influenced by possible sampling biases (e.g. college year and faculties of the students). Thus, it remains to be verified whether the findings in this study can be extrapolated to wider populations.”

[Reviewer’s comment] The researchers always say that the study participation is anonymous, and voluntary, but students can be afraid that they can be identified or their professors can get nervous in general and this can have consequences on the exams.

[Response] Thank you for the important concern. The purpose of the present study and questionnaires were far from the academic disciplines of the students (participants). In our idea, it was not likely that the students were concerned about possible influences of the answer on their grades.

Literature

Al-Shaar.  L., Vercammen, K., Lu, C., Richardson, S., Tamez, M., Mattei, J. (2017): Health Effects and Public Health Concerns of Energy Drink Consumption in the United States: A Mini-Review. Frontiers in Public Health, 5:225.

Attila, S., Cakir, B. (2011): Energy-drink consumption in college students and associated factors. Nutrition, 27, 3, 316–322.

Beacon Health System (2020): Taurine in energy drinks: What is it? www.beaconhealthsystem. org/library/faqs/taurine-in-energy-drinks-what-is-it/ (Accessed: 03.11.2020)

Bedi, N., Dewan, P., Gupta, P. (2014): Energy drinks: potions of illusion. Indian Pediatrics, 51, 7, 529-533.

Breda, J. J., Whiting, S. H., Encarnação, R., Norberg, S., Jones, R., Reinap, M., Jewell, J. (2014): Energy drink consumption in Europe: a review of the risks, adverse health effects, and policy options to Respond. Frontiers in Public Health, 2, 134.

Committee on Nutrition and the Council on Sports Medicine and Fitness (2011): Sports drinks and energy drinks for children and adolescents: are they appropriate? Pediatrics 127, 1182–1189.

EFSA (2015): Scientific Opinion on the safety of caffeine. EFSA Journal, 13, 5, 4102.

Ehlers, A., Marakis, G., Lampen, A., Hirsch-Ernst, K. I. (2019): Risk assessment of energy drinks with focus on cardiovascular parameters and energy drink consumption in Europe. Food and Chemical Toxicology, 130, 109-121.

Kaur, J., Kumar, V., Goyal, A., Tanwar, B., Gat, Y., Rasane, P., Suri, S. (2019): Energy drinks: health effects and consumer safety. Nutrition & Food Science. 49, 6, 1075-1087.

Liauchonak, I., Qorri, B., Dawoud, F., Riat, Y., Szewczuk, M. R. (2019): Non-Nutritive Sweeteners and Their Implications on the Development of Metabolic Syndrome. Nutrients, 11, 3, 644.

Miles-Chan, J. L., Charrière, N., Grasser, E. K., Montani, J. P., Dulloo, A. G. (2015): The thermic effect of sugar-free Red Bull: do the non-caffeine bioactive ingredients in energy drinks play a role? Obesity (Silver Spring), 23, 1, 16-19.

Oddy, W. H., O’Sullivan, T. A. (2009): Energy drinks for children and adolescents. BMJ (British Medical Journal), 339 :b5268

Peacock, A., Droste, N., Pennay, A., Miller, P., Lubman, D. I., Bruno, R. (2015): Awareness of energy drink intake guidelines and associated consumption practices: a cross-sectional study. BMC Public Health, 16, 1, 6.

Persad, L. A. B. (2011): Energy drinks and the neurophysiological impact of caffeine. Frontiers in Neuroscience, 5, 116., 1-8.

Pound, C. M., Blair, B. (2017): Canadian Paediatric Society, Nutrition and Gastroenterology Committee, Ottawa, Ontario. Energy and sports drinks in children and adolescents. Paediatrics & Child Health, 22, 7, 406-410.

Reid, J. L., McCrory, C., White, C. M., Martineau, C., Vanderkooy, P., Fenton, N., Hammond, (2017): Consumption of caffeinated energy drinks among youth and young adults in Canada. Preventive Medicine Reports, 5, 65-70.

Reissig, C. J., Strain, E. C., Griffiths, R. R. (2009): Caffeinated energy drinks - a growing problem. Drug and Alcohol Dependence, 99, 1–3), 1–10.

Seifert, S.M., Schaechter, J. L., Hershorin, E. R. Lipshultz, S. E. (2011): Health effects of energy drinks on children, adolescents, and young adults. Pediatrics, 127, 3, 511-528.

Shah, S. A., Szeto, A. H., Farewell, R., Shek, A., Fan, D., Quach, K. N., Bhattacharyya, M., Elmiari, J., Chan, W., O'Dell, K., Nguyen, N., McGaughey, T. J., Nasir, J. M., Kaul, S. (2019): Impact of High Volume Energy Drink Consumption on Electrocardiographic and Blood Pressure Parameters: A Randomized Trial. Journal of the American Heart Association, 8, 11, e011318.

Subaiea, G. M., Altebainawi, A. F., Alshammari, T. M. (2019): Energy drinks and population health: consumption pattern and adverse effects among Saudi population. BMC Public Health, 19, 1539.

Temple, J. L., Bernard, C., Lipshultz, S. E., Czachor, J. D., Westphal, J. A., Mestre, M. A. (2017): The Safety of Ingested Caffeine: A Comprehensive Review. Frontiers in Psychiatry, 8, Article 80.

van Dam, R. M., Hu, F. B., Willett, W. C. (2020): Coffee, Caffeine, and Health. The New England Journal of Medicine, 383, 369-378.

Visram, S., Hashem, K. (2016): Energy drinks: what’s the evidence. Briefing Paper, Food Research Collaboration, London, 1-12., https://foodresearch.org.uk/publications/energy-drinks/

Round 2

Reviewer 3 Report

The authors made many relevant changes to the manuscript according to the reviewers' comments and suggestions. All comments have been considered. In the revised version of the manuscript, the abstract was rewritten, the Introduction and Discussion was more detailed. The literature citation seems now adequate. Also the English language was revised. Overall, the quality of the manuscript has been much improved.